# Frontiers in Geomorphometry and Earth Surface Dynamics: Possibilities, Limitations and Perspectives

Giulia Sofia[1], John K. Hillier[2], Susan J. Conway[3,4]

5 [1] Department of Land, Environment, Agriculture and Forestry, University of Padova, 35020, Legnaro (PD)

[2] Department of Geography, Loughborough University, Loughborough, LE11 3TU, United Kingdom

[3] Department of Physical Sciences, The Open University, Milton Keynes, MK7 6AA, United Kingdom

[4] Laboratoire de Planétologie et Géodynamique - UMR CNRS 6112, 2 rue de la Houssinière - BP 92208, 44322 Nantes Cedex 3, France

10

*Correspondence to*: Giulia Sofia (giulia.sofia@unipd.it)

**Abstract.** Geomorphometry, the science of quantitative land-surface analysis, has become a flourishing interdisciplinary subject, with applications in numerous fields. The interdisciplinarity of geomorphometry is its greatest strength and also one 15 of its major challenges. Gaps are still present between the process focussed fields (e.g. soil science, glaciology, volcanology) and the technical domain (such as computer science, statistics…) where approaches and theories are developed. Thus, interesting geomorphometric applications struggle to jump between process-specific disciplines, but also struggle to take advantage of advances in computer science and technology. This special issue is therefore focused on facilitating cross-fertilization between disciplines, and highlighting novel technical developments and innovative applications of 20 geomorphometry to various Earth-surface processes. The issue collects a variety of contributions which fall into two main categories: *Perspectives* and *Research,* further divided into 'Research and innovative techniques' and 'Research and innovative applications'. It showcases potentially exciting developments and tools which are the building blocks for the next step-change in the field.

## 1 Introduction

25 Elucidating the dynamics of Earth surface processes through analysis of Digital Elevation Models (DEMs), or 'geomorphometry' (Evans et al., 2003; Hengl and Reuter, 2008; Pike, 1995, 2000), has become a flourishing interdisciplinary subject, with applications in numerous fields (e.g., geomorphology, hydrology, planetary science, archaeology, geo-biology, natural hazards, and computer science). The Earth's morphology can be measured at all scales, from macro (e.g. globally via space missions), to micro (e.g. using laser scanners and most recently structure-from-motion techniques). These datasets have 30 been widely used to analyse both natural (Tarolli, 2014) and anthropogenic (Tarolli and Sofia, 2016) landscapes, and they underpin much modern geomorphological research.

Conceptually any analysis in geomorphometry is a two-step process. Firstly, data must be obtained and their accuracy assessed. Errors are propagated and amplified in surface derivatives (Burrough and McDonnell, 1998; Heuvelink, 1998), thus the usefulness and validity of the results obtained in geomorphometry are intimately associated with the quality of the original

data (Felicísimo, 1994). Secondly, these data are used to derive simplified indices, or are integrated into modelling to portray and understand the specific process of interest. Initially, geomorphometry was used mainly for drainage basin analysis from topographic maps (Miliaresis, 2008), but with time, quantitative techniques and a range of geomorphometric parameters have been developed and applied in an attempt to characterize the landscape and identify processes (Evans, 2012). There are

advantages and disadvantages of each method, technique, parameter and topographic datatype, which vary depending on the objectives of the analysis; often significant weaknesses or methodological limitations exist, which prevent us from gaining the insights into processes that we otherwise might. The interdisciplinarity of geomorphometry is its greatest strength and also one of its major challenges. Specifically, process-focussed fields (e.g. soil science, glaciology, volcanology, hydrology) use their own set of established geomorphometric approaches, and geomorphological specialists often play a key role in developing

these. However, these specialists in turn struggle to incorporate the most innovative approaches and theory being developed in the associated technical domains (such as, computer vision, machine learning, and statistics), or even approaches being used in neighbouring disciplines. So, interesting geomorphometric applications struggle to jump between process-specific disciplines, but also struggle to take advantage of advances in computer science and technology.

If we are to best exploit the wealth of information held within DEMs it is important to i) gather knowledge about the current

technical state-of-the-art in order to consolidate and disseminate established advances; ii) evaluate stubbornly unproductive areas to identify key future challenges and opportunities; iii) provide specific and innovative case studies to assist in cross-disciplinary communication; iv) provide clear and understandable translations from the technical domains where algorithms and techniques find their basis.

In light of the challenges set out above, this special issue in Earth Surface Dynamics highlights current frontiers in

geomorphometry. In order to collect recent research advancements and motivate further research in this direction, we organized a 'Frontiers in geomorphometry' session at the European Geosciences Union General Assembly in 2015, and it has continued successfully since then. The session was focused on facilitating cross-fertilization of best practice across disciplines, highlighting novel technical developments, and showcasing innovative applications of geomorphometry to various Earth-surface processes. The issue collects a variety of contributions, which fall into two main categories: *Perspectives* and *Research,*

where *Research* is further divided into 'Research and innovative techniques' and 'Research and innovative applications' (Table 1).

TABLE 1

The collected *Perspective* works are reviews of state-of-the-art developments as applied to geomorphometry, with a forward-looking component seeking to identify opportunities and challenges. They are intended to stimulate discussion and new

experimental approaches, and they offer a general framework for scientists in different disciplines, dealing with geomorphometry. The papers in the *Research* section present developments of novel techniques, or showcase innovative application(s) of existing methods; the novel techniques are applicable to a variety of dominant geomorphic features, whilst the applications cover different spatial and temporal scales (Figure 1). The works display how geomorphometry can provide

sets of useful techniques and tools for research in different geomorphic and spatio-temporal contexts, given that sufficient data, in sufficient quality, are available.

FIGURE 1

## 2 Frontiers

### 2.1 Perspectives

The collected perspectives investigate three major questions. i) Physical processes, including anthropogenic feedbacks sculpt planetary surfaces (e.g., Earth's). A fundamental tenet of geomorphology is that mapping and, increasingly, quantifying landform features produced can yield insights into the processes. However, the precision and accuracy of mapped data are not well understood. So, how good are these geomorphological data that underpin analyses, and how can we more objectively investigate this? ii) The human brain has a remarkable capability for identifying patterns in complex, noisy datasets, and then applying this knowledge to problem solving. Can we transfer and replicate this ability via computational means, to advance geosciences? iii) One of the most recent revolution in geomorphology is the multiview photogrammetry, or Structure-from-Motion (SfM) technique (Fonstad et al., 2013; Micheletti et al., 2015; Smith et al., 2015; Westoby et al., 2012). What are the key developments and potential future avenues for research in this field, and how do they relate to geomorphometry?

To respond to the first point, Hillier et al. (2015) introduce synthetic DEMs. This perspective reviews the possible approaches to the generation of artificial DEMs. highlighting their limitations, potential, and the opportunities for application. Realistic synthetic DEMs offer a way to assess and understand geomorphological data, allowing users to proceed with uncertainty-aware landscape analysis to examine physical processes.

Valentine and Kalnins (2016) offer an overview about machine learning and its potential in geosciences. Learning algorithms come from the computer science world, and they are designed to replicate the human approach of inferring information from a dataset, and then apply that information predictively. In this work, the authors provide a review of the existing applications in geosciences, and discuss some of the factors that determine whether a learning algorithm approach is suited to geomorphological problems.

Eltner et al. (2016) provide a summary for researchers wanting to apply the SfM method. They summarize the state of the art of published research on SfM photogrammetry applications in geomorphometry. In addition, they give an overview of terms and fields of application, and they identify key future challenges, with a specific focus also on the errors associated with such a technique.

### 2.2 Research and innovative techniques

A fundamental operation in geomorphometry is the extraction of parameters from DEMs to understand the underlying process. How these parameters or objects are evaluated and identified still presents a challenge, and there is still room for improvement.

Papers included here extend our knowledge about sediment dynamics and fluvial incision, or stage-dependent patterns in rivers. A further collection of work focuses on sediment, erosion and connectivity at the hillslope or watershed scale.

Hergarten et al. (2016) develop and explore an extension of the chi-transformation ($\chi$) to small catchment sizes. They solve the limitation of the $\chi$ technique for different watershed sizes, extending the stream power equation to headwater areas dominated by debris flows. In addition, the authors introduce an alternative optimization scheme to linearize the chi-elevation relation.

Brown and Pasternack (2016) demonstrate a relatively new method of analysis for stage-dependent patterns in rivers named geomorphic covariance structures (GCSs). Using meter-scale resolution DEMs, their approach aims to understand if and how the covariance of bed elevation and flow-dependent channel top width are organized in a partially confined, incising gravel-cobble bed river with multiple spatial scales of anthropogenic and natural landform heterogeneity across a range of discharges.

Trevisani and Cavalli (2016) propose a flow-oriented directional measure of surface roughness based on geostatistics that takes into account surface gravity-driven flow directions. Their approach shows the potential impact of considering directionality in the calculation of roughness indices. In addition, they demonstrate how the use of flow-directional roughness can improve the geomorphometric modelling of sediment connectivity, and the interpretation of landscape morphology.

Sklar et al. (2016) propose a novel way to quantify the three-dimensional geometry of catchments. The authors develop an empirical algorithm for generating synthetic source-area power distributions, parameterized with data from natural catchments. Their model can be used to explore the effects of topography on the distribution on fluxes of water, sediment, isotopes and other landscape products passing through catchment outlets.

Bigelow et al. (2016) focus on erosion and sedimentation, and the identification of sediment sources and sinks across landscapes from a practitioners' point of view. Their approach demonstrates a modern analysis of important geomorphic processes affected by land use that can be easily applied by agencies to solve common problems in watersheds, improving the integration between science and environmental management.

Grieve et al. (2016) present software for the automatic extraction and processing of relevant topographic parameters to rapidly generate non-dimensional erosion rate and relief data. This application allows identification of whether landscapes are in topographic steady state, and to identify clear signals of an erosional gradient, or evidence for a landscape decaying following uplift.

## 2.3 Research and innovative applications

In this section, the collected papers expand the applications of geomorphometry to a larger spatial and temporal domain, investigating past tectonic history, or past interactions between ice sheets and climate in glacial systems. Other researchers show the effectiveness of multitemporal datasets at the hillslope or catchment scale to give new insights into sediment dynamics and the seasonal pattern of erosion processes. Finally, two more papers push the frontier of which processes can be examined using SfM for quantitative analysis in the glaciological field.

Andreani and Gloaguen (2016) present a study that uses geomorphic indices to classify the landscape into different regions in order to unravel its tectonic history. These observations/interpretations allow for a better understanding of the recent evolution of the diffuse triple junction between the North American, Caribbean, and Cocos plates in northern Central America.

Wickert (2016) offers a general method to compute past river flow paths, drainage basin geometries, and river discharges at the continental-scale. By integrating numerical modelling (i.e. ice sheet, isostatic adjustment and climate) with field data including geomorphology, his work builds new insights into past glacial systems and climate–ice-sheet interactions.

In Loye et al. (2016), terrestrial Laser Scanning (TLS) is used as a monitoring tool at the catchment scale to analyse the coupling between sediment dynamics and torrent responses in terms of debris flow events. Similarly, Bechet et al. (2015) provide a novel example of how high-resolution time-lapse DEM collection can give insights into processes, in particular for understanding the seasonal pattern of erosion processes for black marls badland-type slopes.

The work by Piermattei et al. (2016) demonstrates the advantages and potential of SfM to calculate the geodetic mass balance of glacier in the Ortles-Cevedale Group, Eastern Italian Alps. In addition, they investigated the feasibility of using the image-based approach for the detection of the surface displacement rate of an active rock glacier. Westoby et al. (2016) analyse the surface evolution of an Antarctic blue-ice moraine using multi-temporal DEMs from TLS and SfM. The authors' results provide an additional understanding of inter-annual development of moraine systems.

## 3 Closing remarks

The availability of DEMs at multiple scales in terms of resolution and spatial and temporal coverage offers great opportunities for the investigation of Earth-surface processes. Geomorphometry has become inter-disciplinary, with focus on new techniques in digital terrain production but also analyses, independent of the subject, and/or field. This special issue showcases exciting developments and tools (e.g. synthetic DEMs, neural networks, Structure-From-Motion) that are the building blocks for the next step-change in the field. Research continues to evolve as computing power increases, and new instrumentation is developed to observe and analyse the Earth and its interacting processes. Geomorphometry is becoming essential to the understanding of global issues, such as natural hazards, sediment production and anthropogenic changes to the Earth system, among others. Such multidisciplinary analytical tools will only become more effective in improving our knowledge of the Earth at a variety of spatio-temporal scales. In reading and compiling the contributions in this Special Issue, we hope that you, the scientific community, will be inspired to seek out collaborations and share your ideas across subject-boundaries, between technique-developers and users, enabling us as a community to fully exploit the wealth of knowledge inherent in our increasingly digital landscape.

**Competing interests:** The authors declare that they have no conflict of interest.

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

**Tables and Table captions**

Table 1: Main themes covered by the research in the Special Issue

| Perspectives | Research and innovative techniques | Research and innovative applications |
|---|---|---|
| *Synthetic DEMs*: | *Sediment dynamics/Fluvial incision*: | *Past tectonic history*: |
| Hillier et al., 2015 | Hergarten et al., 2016 | Andreani and Gloaguen, 2016 |
| *Structure-From-Motion (SfM) Photogrammetry:* | *Stage dependent patterns in rivers*: | *Past interaction between ice sheets and glacial systems*: |
| Eltner et al., 2016 | Brown et al., 2016 | Wickert, 2016 |
| *Learning Algorithms*: | *Erosion and connectivity at the hillslope or catchment scale*: | *Multitemporal dataset to evaluate erosion patterns*: |
| Valentine and Kalnins, 2016 | Sklar et al., 2016; | Loye et al., 2016; |
| | Trevisani and Cavalli, 2016; | Bechet et al., 2016; |
| | Bigelow et al., 2016; | *SfM for glacial processes*: |
| | Grieve et al., 2016 | Westoby et al., 2016; |
| | | Piermattei et al., 2016. |

**Figures and Figure captions**

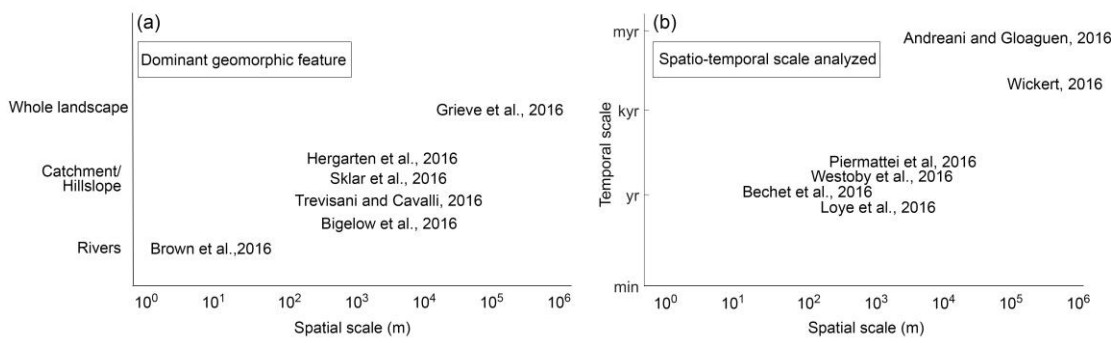

**Figure 1: Dominant geomorphic feature(s) and spatial and temporal scales investigated by the research papers in this special issue:**
10 **(a) dominant geomorphic feature(s) and spatial extent of the suggested applicability of the innovative techniques (Section 2.2)*;* (b) dominant temporal scale and spatial extent covered in upon which the innovative applications are demonstrated (Section 2.3)**