# Peer review of "Frontiers in Geomorphometry and Earth Surface Dynamics: Possibilities, Limitations and Perspectives"

_Earth Surface Dynamics, 2016_

## Referee Comment (RC1) · Anonymous Referee #1 · 13 Jun 2016

The manuscript by Sofia et al. presents an introduction to the special issue entitled "Frontiers in Geomorphometry and Earth Surface Dynamics: Possibilities, Limitations and Perspectives" that collects thirteen contributions in the field of geomorphometry. To this end, the authors provide a convincing overview on the strengths, mainly related to interdisciplinarity and to the wide range of potential fields of application, and major challenges of the science of quantitative land-surface analysis. The contributions are then grouped into Perspective and Research categories and main findings of each article summarized. The manuscript is well presented and suitable for publication.

Minor suggested changes are listed below:

Page 1, L. 26: references: I believe that the geomorphometry book by Hengl and

[Figure]

Reuter is dated 2009 and not 2008. Please check the reference. The work by Pike (1995) could be added to the list since it's one of the first introducing the term "geomorphometry".

Page 1, L. 21: "Research and innovation technique"->"Research and innovative twchnique"

Page 1, L. 25: "...Model"->"...Models"

Page 1, L. 29: Please consider revising the sentence "These datasets have broad applications to all kinds of processes, both natural and anthropogenic..."-> "These datasets have been widely used as a topographic base to analyze both natural and anthropogenic processes..."

Page 1, L.33: you could stress that assessing accuracy of DTM is a very important step in geomorphometry since errors are propagated in DTM derivatives.

Page 1, l. 34: not only modelling but also DTM are widely used to derive more simplified indices or indicators. I think geomorphometric indices could be mentioned here. Few references could be added here.

Page 2, L. 1-2: the sentence seems truncated.

Page 2, L. 29: maybe "landform features" could be used in place of "shape"

Page 3, L. 20: "sediment"->"sediment dynamic"

Page 3, L. 30: "roughness"->"surface roughness"

Page 4, L. 19: "indexes"->"indices"

Cited references

Hengl, T. and Reuter, H. I., Eds.: Geomorphometry: Concepts, Software, Applications, Elsevier., 2009.

Pike, R. J.: Geomorphometry — progress, practice, and prospect, Z. Für Geomorphol.,

Supplementband 101, 221–238, 1995.

**ESurfD**

Interactive
comment

---

## Referee Comment (RC2) · Anonymous Referee #2 · 27 Jun 2016

This is a well-written preface of a Special Issue (SI). The paper introduces the SI on "Frontiers in Geomorphometry and Earth Surface Dynamics: Possibilities, Limitations and Perspectives". I'm pretty sure it will reach a high impact in our Earth science community. I haven't major issues to highlight. Just a suggestion related to the chapt.1 and 3, in addition to other minor comments.

Introduction (chapt.1): Why not providing at the end of the introduction a table or a flow diagram summarizing the papers (and their main findings) collected in this SI? This will be very useful for the readers. Prefaces of SI are always written without any illustrations. Maybe this time should be a good opportunity to provide something more attractive. Note that this is my personal view. In the hands of the authors the decision

if following such advice.

Future challenges (chapt.3): this chapter can be enlarged. I have some difficulties to see specific future challenges here. It seems just a general discussion. I'm sure that the authors have enough background to extract from their articles collection, and from the literature the future challenges of Geomorphometry. Are these only "synthetic DEMs, or neural networks"? Surely not.

Minor comments

Pag. 1, line 27-28: and "hydrology"? also there a DEM can help in supporting analysis, isn't it?

Pag. 1, line 30: switch the order of citation as (Tarolli, 2014; Tarolli and Sofia, 2016), so it is consistent with the previous sequence of words ("natural" first, "anthropogenic" then).

---

## Author Comment (AC1) · 10 Aug 2016

First of all, we wish to thank the two reviewers for their careful evaluation of our manuscript. We have tried our best to address all the issues raised during the review process, and believe that the manuscript benefited from the suggested changes and from further minor editing.

Summary of reviewers' comments.

Reviewer#1 provided some interesting suggestions about enlarging the references in the manuscript, and highlighting the importance of assessing the quality of the DTMs in geomorphometry, especially when dealing with surface derivatives. Other minor

comments involved changes in some words in the text.

Reviewer#2 suggested to provide a flow diagram or a table summarizing the main findings of the articles collected in the special issue, and to enlarge the final chapter about the future challenges.

Below our response to the main point raised, and the related changes to the manuscript.

Please note that the complete rebuttal to each reviewer's comment, and a tracked version of the manuscript are attached as supplement to this comment.

Following Reviewer#1 suggestions, we have connected this work to the wider literature, including some earlier works by Pike (1995,2000), and we provided more references to the importance of DEMs' quality assesment. We also highlighted the fact that DEMs are also used to evaluate indices, or they are integrated into modelling, to portray and understand the specific process of interest.

Following Reviewer#2 comments, we added a new table (table 1) showing an overview of the main themes covered by the research presented in the Special Issue. Furthermore, in the revised text, we provided a new figure (Figure 1) where we illustrate a) the dominant geomorphic feature(s) and spatial extent of the techniques presented in the SI papers, and b) the dominant temporal scale and spatial extent of the applications in the SI papers. We also changed the title of the chapter to 'closing remarks', and we highlighted few more points about the future and challenges of Geomorphometry

Please also note the supplement to this comment:
http://www.earth-surf-dynam-discuss.net/esurf-2016-30/esurf-2016-30-AC1-supplement.pdf

**Supplement:**

**Response to the Reviewer's comments**

On the manuscript **esurf-2016-30** submitted to ESurf

**Manuscript:** Frontiers in Geomorphometry and Earth Surface Dynamics: Possibilities, Limitations and Perspectives

**Authors:**  Giulia Sofia, John K. Hillier, Susan K. Conway

First of all, we wish to thank two reviewers for their careful evaluation of our manuscript. We have tried our best to address all the issues raised during the review process, and believe that the manuscript benefited from the suggested changes and from further minor editing.

Here we provide our detailed answers (regular font) to the reviewers' specific annotations (in italics). Attached is also a version of the manuscript with changes tracked in order to highlight all the changes made.

**Reviewer #1**

*The manuscript by Sofia et al. presents an introduction to the special issue entitled "Frontiers in Geomorphometry and Earth Surface Dynamics: Possibilities, Limitations and Perspectives" that collects thirteen contributions in the field of geomorphometry. To this end, the authors provide a convincing overview on the strengths, mainly related to interdisciplinarity and to the wide range of potential fields of application, and major challenges of the science of quantitative land-surface analysis. The contributions are then grouped into Perspective and Research categories and main findings of each article summarized. The manuscript is well presented and suitable for publication.*

We thank the reviewer for this comment.

*Minor comments:*

*Page 1, L. 26: references: I believe that the geomorphometry book by Hengl and C1 Reuter is dated 2009 and not 2008. Please check the reference.*

Thank you, We have double-checked the date, making sure to use the publisher's website (http://store.elsevier.com/Geomorphometry/isbn-9780123743459/).  This indicates a publication date of late 2008 (i.e. 17th Oct), so we use the reference: Hengl, T., Reuter, H.I. (eds) 2008. Geomorphometry: Concepts, Software, Applications. Developments in Soil Science, vol. 33, Elsevier, 796 pp.

*The work by Pike (1995) could be added to the list since it's one of the first introducing the term "geomorphometry".*

Done; we have added this reference and also a reference to Pike (2000).

*Page 1, L. 21: "Research and innovation technique"->"Research and innovative technique"*

Done; we changed the word.

*Page 1, L. 25: ". . .Model"->". . .Models"*

Done.

*Page 1, L. 29: Please consider revising the sentence "These datasets have broad applications to all kinds of processes, both natural and anthropogenic. . ."-> "These datasets have been widely used as a topographic base to analyze both natural and anthropogenic processes. . ."*

Done.

*Page 1, L.33: you could stress that assessing accuracy of DTM is a very important step in geomorphometry since errors are propagated in DTM derivatives.*

Done, we added a sentence about this ''Errors are propagated and amplified in surface derivatives (Burrough and McDonnell, 1998; Heuvelink, 1998), thus the usefulness and validity of the results obtained in geomorphometry are intimately associated with the quality of the original data (Felicísimo, 1994)."

*Page 1, l. 34: not only modelling but also DTM are widely used to derive more simplified indices or indicators. I think geomorphometric indices could be mentioned here. Few references could be added here.*

We thank the reviewer for the suggestion. We added few sentences in this part of the manuscript. We believe that for the purpose of the editorial, adding references to specific indices or parameters (or models) without detailed explanations, would add complexity. We'd rather keep the text more clean and general, thus we opted for a simple approach, as follows 'Second, these data are used to derive simplified indices, or are integrated into modelling, to portray and understand the specific process of interest. Initially, geomorphometry was used mainly for drainage basin analysis from topographic maps (Miliaresis, 2008), but with time, quantitative techniques, and various geomorphometric parameters have been developed and applied in an attempt to characterize the landscape and identify processes (e.g. Evans, 2012).''

*Page 2, L. 1-2: the sentence seems truncated.*

We think the sentence is correct, but to emphasise its link to the previous sentence we now use a semi-colon to link the two.

*Page 2, L. 29: maybe "landform features" could be used in place of "shape"*

Done; we have changed the term.

*Page 3, L. 20: "sediment"->"sediment dynamic"*

Done; we have changed the term.

*Page 3, L. 30: "roughness"->"surface roughness"*

Done; we have changed the term.

*Page 4, L. 19: "indexes"->"indices"*

Done; we have changed the term.

**Reviewer #2**

*This is a well-written preface of a Special Issue (SI). The paper introduces the SI on "Frontiers in Geomorphometry and Earth Surface Dynamics: Possibilities, Limitations and Perspectives". I'm pretty sure it will reach a high impact in our Earth science community. I haven't major issues to highlight. Just a suggestion related to the chapt.1 and 3, in addition to other minor comments.*

We thank the reviewer for his/her comments.

*Introduction (chapt.1): Why not providing at the end of the introduction a table or a flow diagram summarizing the papers (and their main findings) collected in this SI? This will be very useful for the readers. Prefaces of SI are always written without any illustrations. Maybe this time should be a good opportunity to provide something more attractive. Note that this is my personal view. In the hands of the authors the decision if following such advice.*

We thank the reviewer for his/her comments. In the revised text we added a new table (table 1) showing an overview of the main themes covered by the research presented in the Special Issue. Furthermore, in the revised text, we took inspiration from the figure below by (Anderson and Burt, 1990), and we adapted the same concept to a new figure (now Figure 1 in the revised manuscript) where we illustrate a) the dominant geomorphic feature(s) and spatial extent of the techniques presented in the SI papers, and b) the dominant temporal scale and spatial extent of the applications in the SI papers.

[Figure]

Figure 1: Scales in hydrology and geomorphology. The figure shows in a simple way some dominant features of each discipline in a spatial and spatio-temporal context (Anderson and Burt, 1990).

*Future challenges (chapt.3): this chapter can be enlarged. I have some difficulties to see specific future challenges here. It seems just a general discussion. I'm sure that the authors have enough background to extract from their articles collection, and from the literature the future challenges of Geomorphometry. Are these only "synthetic DEMs, or neural networks"? Surely not.*

We thank the reviewer for this comment. We changed the title of the chapter to 'closing remarks', and we highlighted few more points about the future and challenges of Geomorphometry

*Minor comments*

*Pag. 1, line 27-28: and "hydrology"? also there a DEM can help in supporting analysis, isn't it?*

We have added the term.

*Pag. 1, line 30: switch the order of citation as (Tarolli, 2014; Tarolli and Sofia, 2016), so it is consistent with the previous sequence of words ("natural" first, "anthropogenic" then).*

Done.

**References**

Anderson, M.G., Burt, T.P., 1990. Process studies in hillslope hydrology: an overview, in: Anderson, M.G., Burt, T.P. (Eds.), Process Studies in Hillslope Hydrology. Wiley, pp. 1–8.

[revised manuscript text omitted]